# Ellagic Acid–Cyclodextrin Complexes for the Treatment of Oral Candidiasis

**DOI:** 10.3390/molecules26020505

**Published:** 2021-01-19

**Authors:** Aline da Graça Sampaio, Aline Vidal Lacerda Gontijo, Gabriela de Morais Gouvêa Lima, Maria Alcionéia Carvalho de Oliveira, Laura Soares Souto Lepesqueur, Cristiane Yumi Koga-Ito

**Affiliations:** 1Oral Biopathology Graduate Program, São José dos Campos Institute of Science & Technology, São Paulo State University, UNESP, São Paulo 12245-000, Brazil; aline.sampaio@unesp.br (A.d.G.S.); aline.gontijo@gmail.com (A.V.L.G.); gabrielademorais@yahoo.com.br (G.d.M.G.L.); macoliveira12@gmail.com (M.A.C.d.O.); lalepesqueur@hotmail.com (L.S.S.L.); 2School of Dentistry, Santo Amaro University, São Paulo 04743-030, Brazil; 3Department of Environment Engineering, São José dos Campos Institute of Science & Technology, São Paulo State University, UNESP, São Paulo 12247-016, Brazil

**Keywords:** *Candida albicans*, oral candidiasis, ellagic acid, cyclodextrin

## Abstract

The increase in the prevalence of fungal infections worldwide and the rise in the occurrence of antifungal resistance suggest that new research to discover antifungal molecules is needed. The aim of this study was to evaluate the potential use of ellagic acid–cyclodextrin complexes (EA/HP-β-CD) for the treatment of oral candidiasis. First, the effect of EA/HP-β-CD on *C. albicans* planktonic cells and biofilms was evaluated. Then, the cytotoxicity of the effective concentration was studied to ensure safety of in vivo testing. Finally, the in vivo effectiveness was determined by using a murine model of induced oral candidiasis. Data was statistically analyzed. The minimal inhibitory concentration of EA/HP-β-CD was 25 µg/mL and a concentration of 10 times MIC (250 µg/mL) showed an inhibitory effect on *C. albicans* 48 h-biofilms. The complex at concentration 250 µg/mL was classified as slightly cytotoxic. In vivo experiments showed a reduction in fungal epithelial invasion after treatment with EA/HP-β-CD for 24 h and 96 h when compared to the negative control. In conclusion, the results demonstrated that EA/HP-β-CD has antifungal and anti-inflammatory effects, reducing the invasive capacity of *C. albicans*, which suggests that EA/HP-β-CD may be a promising alternative for the treatment of oral candidiasis.

## 1. Introduction

Oral candidiasis affects 12 to 24.2% of immunocompromised patients and 42 to 66.7% of these cases are attributed to *Candida albicans* [1,2]. These infections may compromise the life of immunosuppressed patients, as the oral cavity may be a portal of entrance to systemic infections. [3]. Candidemia is related to high rates of morbidity and mortality (21.7 to 58%) [4,5,6] and these rates can be even higher among patients admitted to an Intensive Care Unit [7].

Conventional antifungal drugs have several limitations such as limited spectrum of activity, interaction with other drugs, high cost, and toxicity [8]. Additionally, the increase in the occurrence of antifungal resistance is an important challenge [9,10,11]. Associated with this, it is also known that *Candida* biofilms are difficult to eradicate and they are frequently responsible for therapeutic failures [10,12,13]. Due to these limitations, the number of therapeutic options is very low and few molecules are used in clinical practice [14]. Thus, research for new therapies and discovery of new antifungal molecules are needed [15,16,17].

New antifungal molecules from Brazilian Pantanal plants have been reported [18,19,20,21]. Among these molecules, ellagic acid showed promising activity against *Candida albicans* [19].

Ellagic acid can be found in herbs, fruits, and vegetables, such as raspberries, strawberries, and walnuts [22]. These plants produce ellagic acid in the form of hydrolyzed tannins, known as ellagitannins, that present antioxidant, antimutagenic, antimicrobial, and anti-inflammatory actions [23,24,25,26,27]. Promsong et al. [28] suggested that this natural polyphenol can help the regeneration and improvement of mucosal innate immunity, preventing superficial infections.

Some characteristics of ellagic acid, such as the low solubility in water, limited permeability, and first pass effect can interfere with bioavailability [19,24,29,30]. Thus, solutions to overcome these problems and improve the pharmaceutic delivery of this molecule have been proposed [30]. One of these strategies is the inclusion of cyclodextrins. Cyclodextrins have been used to improve the solubility of drugs in water [31] and to increase the bioavailability [32,33,34,35]. They are oligosaccharide molecules and macrocycle carriers that are able to form soluble complexes with lipophilic drugs [31].

The viability of using 2-hydroxypropyl-beta-cyclodextrin (HP-β-CD) as a carrier for ellagic acid (EA), with improvement in water solubility and dissolution, has been previously reported [29,36,37]. The investigation on the biological effects of ellagic acid/HP-β-CD (EA/HP-β-CD) complex has given positive results. EA/HP-β-CD showed improved anti-inflammatory activity in vitro by increasing the protection of the erythrocyte membrane from denaturation and lysis [36]. This effect was confirmed in an in vivo study where EA/HP-β-CD reduced significantly the edema induced by lambda carrageenan in rats [38]. Besides, EA/HP-β-CD also showed anti-arthritis in rats, reducing the expression of pro-inflammatory cytokines [39]. Regarding antimicrobial activity, it was reported that the inclusion of EA in HP-β-CD did not interfere with the inhibitory effect against *Candida albicans*, *Escherichia coli*, *Pseudomonas aeruginosa*, *Bacillus luteus,* and *Listeria monocytogenes* [29,37].

Recently, it was reported that EA/HP-β-CD reduced candida invasion in an in vitro invasive candidiasis model [29]. However, the in vivo effect of the complex has not been studied so far. Thus, the aim of this study was to evaluate the application of EA/HP-β-CD for the treatment of oral candidiasis in a murine model. To our knowledge, this is the first comprehensive study of the effects of EA/HP- β-CD on *Candida albicans* biofilms, its toxicity, and application for the treatment of experimentally induced oral candidiasis.

## 2. Results

### 2.1. Determination of MIC of EA and EA/HP-β-CD Complex

The minimal inhibitory concentrations (MIC) of EA and EA/HP-β-CD complex against *C. albicans* ATCC 18804 were 25 µg/mL.

### 2.2. Effect of EA/HP-β-CD on C. albicans Pre-Formed Biofilms

*C. albicans* ATCC 18804 24 h-biofilms were not inhibited by both EA and EA/HP-β-CD. Statistically significant reduction in viability was only observed after treatment of 48 h-biofilms with EA/HP-β-CD compared with the control and pure EA (Figure 1).

### 2.3. Toxicity Analyses

#### Cytotoxicity of Ellagic Acid and EA/HP-β-CD

The toxicity of ellagic acid and EA/HP-β-CD to fibroblast 3T3 was evaluated by MTT bioassay and the percentages of cell viability were classified according to Sletten and Dahl [40] (Figure 2). The testing concentrations were 25 µg/mL (MIC), 50 µg/mL (2 times MIC), 100 µg/mL (4 times MIC), and 250 µg/mL (10 times MIC).

Non-complexed ellagic acid was classified as slightly cytotoxic at concentrations of 25 µg/mL and 100 µg/mL (91, 79 and 81% cell viability, respectively). The concentration of 250 µg/mL was classified as moderately cytotoxic, with 59% cell viability.

After complexation, the cell viabilities for 25 µg/mL, 50 µg/mL, and 100 µg/mL, were, respectively, 77.9, 67.3, and 67.3 and were classified as slightly cytotoxic. For EA/HP-β-CD at 250 µg/mL the cell viability was 63% and was thus also classified as slightly cytotoxic.

### 2.4. In Vivo Effect of EA/HP-β-CD in the Treatment of Oral Candidiasis Experimentally Induced in Murine Model

Complex EA/HP-β-CD containing 250 µg/mL of EA was used for in vivo experiments based on the antibiofilm effect on *C. albicans*. The clinical diagnosis of oral candidiasis was carried out by macroscopic analysis, and then a treatment protocol was started. Animals were treated for 3 sequential days and euthanized after 24 h and 96 h from the last treatment.

The number of viable fungal cells was quantified in the specimens collected (Figure 3). It was observed that when animals were euthanized 24 h after the last treatment, counts of *C. albicans* were 4.5 × 10^4^ ± 4.9 × 10^4^, 1.5 × 10^3^ ± 1.7 × 10^3^, and 1.0 × 10^4^ ± 6.4 × 10^3^ CFU/mL for control, nystatin, and EA/HP-β-CD groups. Significant differences were detected among the treatment groups (nystatin and EA/HP-β-CD) and the control group (*p* < 0.05).

No significant differences were observed among the fungal counts obtained for animals euthanized 96 h after the last treatment (control—3.3 × 10^3^ ± 1.7 × 10^3^, nystatin—1.2 × 10^4^ ± 2.0 × 10^4^, and EA—4.6 × 10^3^ ± 5.5 × 10^3^ CFU/mL).

Subsequently, the animal’s tongues were analyzed histologically, by microscopic observation using 200× and 400× magnification.

Morphological analyses of Hematoxylin–Eosin (HE) (Figure 4) from the control group euthanized 24 h after treatment, showed hyperkeratinization of the stratified epithelium with basal layer disorganization, spongiosis, and exocytosis. The presence of lymphocytes and plasma cells, rare polymorphonuclear cells, and intense inflammatory infiltrate could be observed in the connective tissue.

The complex EA/HP-β-CD 24 h group exhibited inflammatory characteristics, less severe than those observed for the control group in general, but with a remarkable presence of epithelial micro-abscesses. Hyperkeratinized epithelium with preserved lingual papillae was observed. Epithelial cells did not show cell inflammatory alterations. Next to the basal layer, some areas of spongiosis could be observed. Duplication of basal layer could be seen with sparse areas of exocytosis. In the subepithelial region, increased presence of plasma cells was noted. Moderate mononuclear inflammatory infiltrate was present in the connective tissue.

In the Nystatin 24-h group, mild inflammatory alterations could be noted in a way in that the original histological characteristics were preserved. Animals showed acanthosis, alteration on the basal layer, exocytosis, and loss of papillae in candidiasis areas. In the same region mild inflammatory infiltrate was composed of lymphocytes, plasma cells, and neutrophils.

In all groups analyzed, despite the manifestations of oral candidiasis, muscular, adipose and glandular tissues were preserved. Inflammatory alterations of the 24-h groups are illustrated in Figure 4.

The control group euthanized 96 h after treatment showed histopathologic signs of oral candidiasis. Hyperkeratinized epithelium with loss of tongue papillae and basal layer duplication could be detected. Intense mononuclear inflammatory infiltrate was observed in deeper connective tissue areas.

Tongue fragments of the HP-β-CD 96 h group showed similar findings when compared to HP-β-CD 24 h. Mild characteristics related to candidiasis were detected. Less inflammatory alterations were noted. Some intra-epithelial abscesses were observed but they were small and limited to the epithelial surface. Histopathologic characteristics between the nystatin 96 h group were similar to that described for the nystatin 24 h group. Histological sections from the 96 h groups can be seen in Figure 5.

The semi quantitative analyses showed no statistical significance among the groups (*p* > 0.05; Kruskal–Wallis, following Dunn’s multiple comparison test), for both periods, shown in Figure 6c,d. Although it was not statistically different, EA/HP-β-CD (median = 2) for both periods and the gold standard nystatin (median = 1 in the 24 h and 2.5 in 96 h) provided minor inflammatory alterations compared to the control group (median = 2.5 in 24 h and 3.5 in 96 h). Figure 6a,b shows the frequency distribution for inflammatory scores.

PAS analyses after 24 h of the last treatment showed that complex EA/HP-β-CD group (median = 0) could reduce tissue invasion by *Candida* hyphae and there was a significant difference in the amount of *Candida* hyphae invading the tissue when compared to negative control (median = 2) (Figure 7). The nystatin group (median = 1) also showed significant difference and lower invasion when compared to negative control. (*p* < 0.0001; Kruskal-Wallis, followed Dunn’s multiple comparison test), in shown in Figure 8. After 96 h of the last treatment showed significant difference of the control (median = 1) between the complex EA/HP-β-CD group and nystatin group. In the nystatin group and complex EA/HP-β-CD the score median was = 0.

## 3. Discussion

Previous studies indicated a promising inhibitory action of EA on *Candida albicans* [19]. Gontijo et al. [29] evaluated the action of ellagic acid complexed with cyclodextrin in an in vitro invasive candidiasis model and obtained a reduction of fungal invasion. *Candida*’s resistance to conventional antifungals such as azoles. polyenes, and echinocandins, are recurrent, requiring the search for new therapeutic treatments against this fungus [41,42,43]. However, there are no data on the in vivo effect of this molecule for the treatment of oral candidiasis.

As reported in the literature, EA has poor aqueous solubility [23,24,37,44] and the cell membrane has low permeability to EA [37], which can compromise its bioavailability. Cyclodextrin was thus used in order to improve its solubility. The cyclodextrin used in the present study, hydroxypropyl beta-cyclodextrin (HP-β-CD), was selected because it is one of the most used and has the ability to provide better complexation results according to the literature [32]. The HP-β-CD molecule has hydroxypropyl groups in its formula that can result in a large increase in solubility of up to 60% mass by volume (*m*/*v*), compared to other cyclodextrins, such as β-CD, which usually produce insoluble or low complex solubility [32,45,46,47]. The complexation process and characterization of the complex were described in a previous study [29].

The antibiofilm action of the isolated molecule and the EA/HP-β-CD complex was initially evaluated. The concentration 10 times MIC (250 µg/mL) was chosen since biofilms are more resistant to antimicrobial agents than planktonic cells. Thus, higher concentrations than MIC are usually necessary to be more effective against *Candida* biofilms, as previously described in the literature [48,49]. Therefore, concentrations lower or higher than 10 times MIC (250 µg/mL) would probably be, respectively, less effective and toxic or insoluble in culture medium. In these experiments, it was found that EA/HP-β-CD at a concentration of 250 µg/mL had an effect against a biofilm of 48 h. However, this result was not observed for 24-h biofilms. Since biofilm maturation occurs between 24 and 48 h, forming a more complex and increased multi-layer matrix with all fungal cells present [50], the greater activity in mature biofilms was unexpected. This result is similar to that reported by Teodoro et al. [47], using gallic acid. It is known that the properties of a biofilm formed in 24 h differ from a biofilm in 48 h, such as the molecular characteristics and water content [51]. Possibly, these differences associated with the different components of the biofilm matrix may be associated with the better performance of the compounds on biofilms of 48 h [52]. The distribution of the drug is aided by the presence of water channels in the structure of the biofilm and the components of the biofilm matrix can delay this access [52]. Drug diffusion rates may be distinct when comparing biofilms formed by different strains of *C. albicans*, which is further evidence that the effectiveness of a molecule can be influenced by many factors related to the specificities of each biofilm [52].

A cytotoxicity test was carried out in order to evaluate the cell viability as a function of different concentrations. The effect on fibroblasts was investigated. Pure and complexed ellagic acid demonstrated moderate and slight cytotoxicity, respectively, at the highest tested concentration of 250 µg/mL, according to a previous study [40], showing cell viability of 58 to 63%, respectively. The results corroborate the study by Lourenção Brighenti et al. [19], in which non-complexed ellagic acid also did not show cytotoxicity, with 80% cell viability after 24 h of exposure to a concentration of 50 µg/mL. In the present study, at the same concentration (50 µg/mL), a similar value of 79.4% of cell viability was found in the pure substance and 67.3% in the substance containing HP-β-CD. Cytotoxicity tests on different cell lines, such as rat embryonic fibroblast (NIH/ 3T3) and Madin–Darky bovine kidney (MDBK) and human foreskin fibroblast (HFF) showed that cell viabilities were equal or more than 50% at concentrations equal or higher than 790 µM (238.6 µg/mL) after 24 h of exposure for EA complexed in β-CD [53]. Furthermore, the complexation with β-CD did not affect cell viability [53], as observed in this present study. A similar finding was observed by Weisburg et al., [54] that did not observe cytotoxicity in normal human gingival fibroblasts (HF-1) at a concentration of 200 µM (60.4 µg/mL) after 24 h of exposure. Although further studies should be carried out to determine the chronic toxicity of EA/HP-β-CD, these sets of findings suggest the safe use of EA following its administration, even at higher concentrations, which is a potential advantage for its use on biofilms. Indeed, the required concentration to disrupt biofilms could be several times the MIC [48]. Therefore, further experiments were performed with concentrations 10 times the MIC (250 µg/mL).

Owing to the low cytotoxicity and the promising antibiofilm activity of EA/HP-β-CD complex containing 250 µg/mL of EA, an in vivo study with mice was performed with this new formulation at this concentration to evaluate its effect for the treatment of oral candidiasis. The application of this new formulation was selected since candidiasis is the most common infection in oral mucosa, especially in immunocompromised patients [55]. Additionally, topic administration of this formulation has an advantage from a biopharmaceutical point of view, in that as EA crosses the membrane slowly [29], the effect will be probably limited to a local tissue, which would reduce possible systemic side effects.

The methodology of inducing oral candidiasis in mice proved to be effective, validated by the characteristic lesions on the tongue and corroborated by the observation of hyphae and yeasts in the histopathological analysis. It is important to highlight that the occurrence of a disease is the involvement of the complex interaction between the microorganism and the host, involving the expression of *Candida* virulence factors, the interactions of the bacterial microbiome and the host’s own immune system [56]. Thus, in vivo model studies become important for the validation of results obtained in vitro and for the generation of data to support future clinical use.

Similarly, to the reduction of viable cells in a 48-h biofilm treated with EA/HP-β-CD detected in vitro, a significant reduction in the colony forming unit (CFU) count of *C. albicans* was observed in the tongues of animals with experimentally induced oral candidiasis after 24 h of the last treatment, but not significant for the tissue inflammatory response.

In the present project, EA/HP-β-CD was applied topically, twice a day, for three days. This multidose regimen is recommended for nystatin, considered the gold standard for the treatment of oral candidiasis [42]. It should be noted that treatment with nystatin for oral candidiasis can vary from 1 to 4 weeks and there is no consensus on the formulation, dosage, or duration of treatment [57]. Quindós et al. [58] highlighted that the Infectious Diseases Society of America (IDSA) recommends nystatin in doses ranging from 4–6 mL (liquid drug) four times a day, or the administration of 1–2 lozenges, four times a day for 7–14 days for the treatment of mild oral candidiasis. Thus, the dosage of nystatin can vary between 100,000 U to 200,000 U. However, Lyu et al. [57], demonstrated that nystatin at a dosage of 400,000 U has better effects on fungi. Thus, future studies, increasing the number of treatment days, could be performed to seek a better response to nystatin and EA/HP-β-CD. Besides, the association of EA/HP-β-CD and other conventional antifungals should be investigated.

The histopathological analyses of the animals’ tongues submitted to the induction of experimental oral candidiasis indicated a reduction in the tissue invasion of hyphae of *C. albicans* to the epithelium, in the groups exposed to the treatments of ellagic acid complexed in HP-β-CD and nystatin, after both treatment times, when compared to the non-treated control.

The groups treated with EA complexed in cyclodextrin, reached similar levels of tissue invasion in relation to the group treated with nystatin, after 24 h. After 96 h, the group treated with nystatin showed the highest reduction when compared to the EA/HP-β-CD group. The histopathologic analyses of the treated groups showed less inflammatory alterations and better conservation of the tissue characteristics than the control group. These effects were more evident after 96 h of treatment, although the median values of the histological scores were not statistically significant. These findings corroborate previous in vitro studies that reported the anti-inflammatory and antimicrobial effect of EA treatment [23,24,25].

Oral topical application of EA/HP-β-CD reduced fungal tissue invasion and had an anti-inflammatory effect. These outcomes suggest that the complex has the potential to be used clinically for the treatment of oral candidiasis, particularly in refractory cases and when there is resistance to conventional antifungals. Future investigations on other therapeutic regimens and association with conventional antifungals may reveal new potentials for the clinical use of EA/HP-β-CD.

## 4. Materials and Methods

### 4.1. EA/HP-β-CD Complex Formation and Determination of Minimal Inhibitory Concentration (MIC) of EA and EA/HP-β-CD

The complexation of ellagic acid and characterization of EA/HP-β-CD were previously described by Gontijo [29]. The minimal inhibitory concentration (MIC) of pure EA and EA/HP-β-CD complex were determined for *Candida albicans* ATCC 18804. The MIC assessment was performed according to the microdilution technique describe in the M27-A3 standard document by Clinical Laboratory Standards Institute (CLSI) [59]. The inoculum was standardized spectrophotometrically (λ: 530 nm and O.D.: 0.138), obtaining approximately 10^6^ cells/mL. This suspension was diluted in Roswell Park Memorial Institute medium (RPMI) 160 broth (pH 7.0) to obtain a concentration of 10^3^ cells/mL. EA concentrations between 50 and 0.08 µg/mL were tested. The plates were incubated for 24 h at 37 °C, in aerobiosis. Fungal growth was analyzed visually and compared to the positive control. The MIC was determined as the concentration inhibiting 80% of growth compared to the control.

### 4.2. Effect of EA/HP-β-CD on C. albicans Pre-Formed Biofilms

The antibiofilm effect of EA/HP-β-CD was evaluated according to the methodology of Cheng et al. [60], with modifications. EA/HP-β-CD (10 times MIC, 250 µg/mL) and pure EA (250 µg/mL) were tested. Briefly, standardized *C. albicans* ATCC 18804 suspension in a concentration of 10^7^ cells/mL was obtained spectrophotometrically. Aliquots of 20 μL of the fungal inoculum was added to 180 μL RPMI broth supplemented with glucose 2% in 96 well plates. Plates were incubated for 90 min at 37 °C, under shaking (80 rpm), for the pre-adhesion phase. After this period, each well was washed with sterile saline solution (NaCl 0.9%) and fresh culture medium was added. Biofilms of 24 and 48 h were obtained. For 48 h biofilms, the culture medium was refreshed after 24 h incubation.

Biofilms were exposed to EA/HP-β-CD (250 µg/mL) and pure EA (250 µg/mL) for 1 min at room temperature. Non-exposed growth control was included for comparative purposes. Then, the biofilms were washed with sterile physiologic solution (NaCl 0.9%). The biofilms were dispersed, serially diluted, and plated on Sabouraud dextrose agar. Plates were incubated at 37 °C for 24 h. After the incubation period, the number of colony forming units (CFU) per biofilm was obtained. The tests were performed in triplicate on 3 different occasions.

### 4.3. Toxicity Analyzes

#### Cytotoxicity of Ellagic Acid and EA/HP-β-CD

Fibroblasts 3T3 were cultivated in Dulbecco modified Eagle medium (DMEM) supplemented with 10% inactivated bovine fetal serum and penicillin/streptomycin 1% and maintained at 37 °C and 5% CO_2_. The cells (6 × 10^4^ cells/well, passages 13 and 17) were transferred to 96-well microplates and incubated at 37 °C, 5% CO_2_ for 24 h. Then, the culture medium was removed and the cells were treated with pure EA or EA/HP-β-CD complex, previously solubilized in DMEM at different concentrations, from 25 to 250 μg/mL. The microplates were incubated at 37 °C, 5% CO_2_ for 24 h. Cell bioactivity was measured by MTT according to Mosmann [61]. Non-treated cells and DMEM medium were used as controls. The bioactivity was determined by comparing the percentage of cell viability, considering the non-treated group as 100%. The assays were performed in triplicate on 2 different occasions. The treatment was classified according to the final cell viability as severely cytotoxic (less than 30%), moderately cytotoxic (between 30 and 60%), and slightly cytotoxic (greater than 60%), according to Sletten and Dahl [40].

### 4.4. In Vivo Effect of EA/HP-β-CD in the Treatment of Oral Candidiasis Experimentally Induced in Murine Model Structures

The methodology of experimentally induced oral candidiasis was based on Okada [62] and Borges [63], with modifications. 

Thirty male mice (Mus musculus, Swiss), with 55 days and approximately 45 g of weight were randomly divided into 3 groups, according to the treatment: (i) Complex EA/HP-β-CD containing 250 µg/mL of EA; (ii) Positive control—Nystatin solution 100,000 UI/mL (Nystatin, Neoquímica, Rio de Janeiro, Brazil); and (iii) Negative control—PBS phosphate buffered saline (pH 7.4).

Animals were immunosuppressed with prednisolone (100 mg/kg Depro-Medrol, Pfizer, Belgium), each 48 h for up to 5 days. A solution of tetracycline (0.83 mg/mL; Terracimina, Zoetis, Guarulhos, São Paulo, Brazil) was added to the drinking water in all the experimental periods. Treatment with prednisolone and tetracycline started one day before the experimental infection. Animals were maintained at controlled environmental conditions (temperature at 20 °C and 12 h light cycle in ventilated racks and received water and food ad libitum).

*C. albicans* ATCC 18804 standardized suspensions (10^8^ cells/mL) were prepared in a spectrophotometer (λ: 530 nm and O.D.: 1.258). Animals were anesthetized with 10% ketamin (Dopalen, 0.2 mL/100 g body weight; Ceva, Paulínia, Brazil) and 2% xylazine (Anasedan, 0.1 mL/100 g body weight; Ceva, Brazil). The animals were inoculated with sterile swabs soaked with fungal suspension for 5 min in contact with the mice’s tongue. Inoculation was performed on days 2 and 4 of the experiment. The diagnosis of candidiasis was done 48 h after the last inoculation. Animals with clinical diagnosed lesions were divided into the experimental groups (*n* = 10 each group).

Treatments were administered twice a day (8 a.m. and 2 p.m.), intra-orally using micropipettes, for 3 days in non-anesthetized animals. After 24 and 96 h from the last treatment, animals were submitted to euthanasia with excessive dose of anesthetic. Tongues were removed and microbiologic and histological analyzes were performed. The experimental arrangement is illustrated in Figure 9.

Six animals of each group were randomly selected for microbiologic analyses of the tongue. Tongues were removed, macerated, and treated with a solution of 0.25% trypsin for 20 min, according to [64]. The final suspension was plated in Sabouraud dextrose agar supplemented with chloramphenicol and for CFU/tongue values were calculated.

Tongues from 4 animals per group were prepared for histopathological analyses. After being surgically removed, the tongues were fixed in 4% paraformaldehyde solution, processed, and embedded in paraffin. Slices of 5 microns were obtained and stained with PAS (Periodic acid-Schiff) and HE (Hematoxylin-Eosin), aiming to observe fungal invasion and tissue alterations, respectively. All the analyses were performed with optical microscopy by a single examiner.

For PAS analyzes, two slices per animal were analyzed, for a total of 42 fields per tongue. The number of invading hyphae was counted in all fields and the counts were scored: 0—absence of hyphae; 1—1 to 5 hyphae; 2—6-15 hyphae; 3—16 to 50 hyphae; and 4—more than 50 hyphae [65].

For HE analyzes, the slices were observed in distinct fields at 200x and 400x magnification (Axioskop 40, Carl Zeiss, Oberkochen, Germany) for descriptive analyses of the tissue inflammatory response. Scores were attributed according to the number and type of inflammatory epithelial alterations (6 fields/tongue). Presence of hyperkeratosis, hyperplasia, acanthosis, exocytosis, spongiosis, alteration in the basal layer pattern, loss of lingual papillae, and presence of intraepithelial micro abscess were assessed and scores were attributed according to Table 1.

### 4.5. Data Analyses

All the experiments were previously analyzed by normality test followed by suitable statistical analysis. Statistical analyses and graphs were performed using GraphPad Prism, version 7.0 (Graphpad Software Inc., San Diego, CA, USA). The results of antibiofilm effects were compared by One-way ANOVA followed by Newman–Keuls multiple comparison post-hoc test. The values of cytotoxicity were obtained by percentage analysis. For the in vivo assays, microbiological data was analyzed with One-way ANOVA test, followed by Newman–Keuls multiple comparison post-hoc test. Histological (PAS and HE) scores (median values) were analyzed by Kruskal–Wallis with Dunn’s post hoc tests. The significance level was determined as 5%.

## 5. Conclusions

In conclusion, the results demonstrated that the EA/HP-β-CD has antifungal and anti-inflammatory effects, showing a reduction in the invasive capacity of *C. albicans,* which suggests that EA/HP-β-CD may be a promising alternative for the treatment of oral candidiasis.

## Figures and Tables

**Figure 1 molecules-26-00505-f001:**
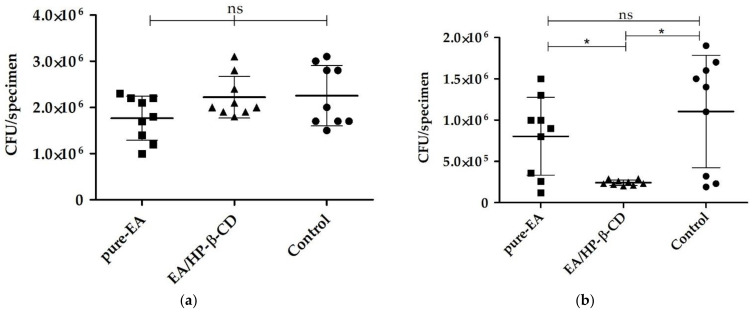
Effect of ellagic acid (EA) and EA/HP-β-CD on *Candida albicans* ATCC 18804. (**a**) 24 h-biofilms; (**b**) 48 h-biofilms. Viability after exposition to 250 µg/mL (10 times MIC) for 1 min. Data was analyzed by ANOVA and Newman–Keuls multiple comparison test (n.s. = not significant; * *p* < 0.05).

**Figure 2 molecules-26-00505-f002:**
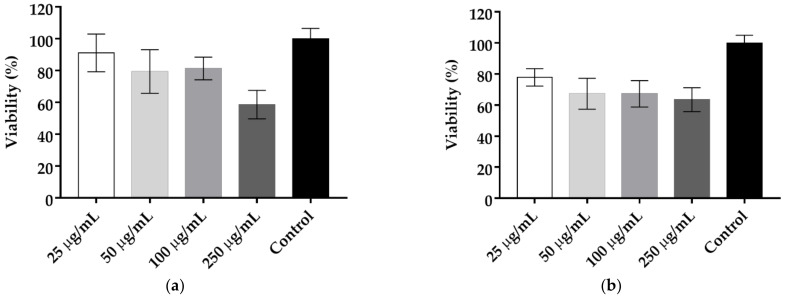
Viability of fibroblast 3T3 after 24 h exposition to 25 µg/mL, 50 µg/mL, 100 µg/mL, and 250 µg/mL of: (**a**) Non-complexed ellagic acid; (**b**) Complexed EA/HP-β-CD.

**Figure 3 molecules-26-00505-f003:**
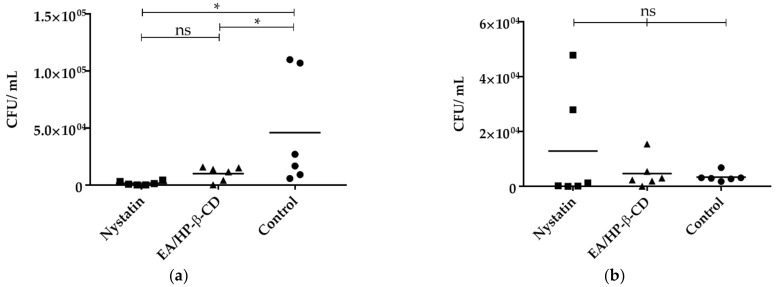
Number of viable fungal cells (CFU/mL) recovered: (**a**) 24 h; (**b**) 96 h; after the end of the treatment with nystatin and complex EA/HP-β-CD in relation to negative control. Statistical test ANOVA followed by Newman-Keuls test. * *p* < 0.05. ns—not significant.

**Figure 4 molecules-26-00505-f004:**
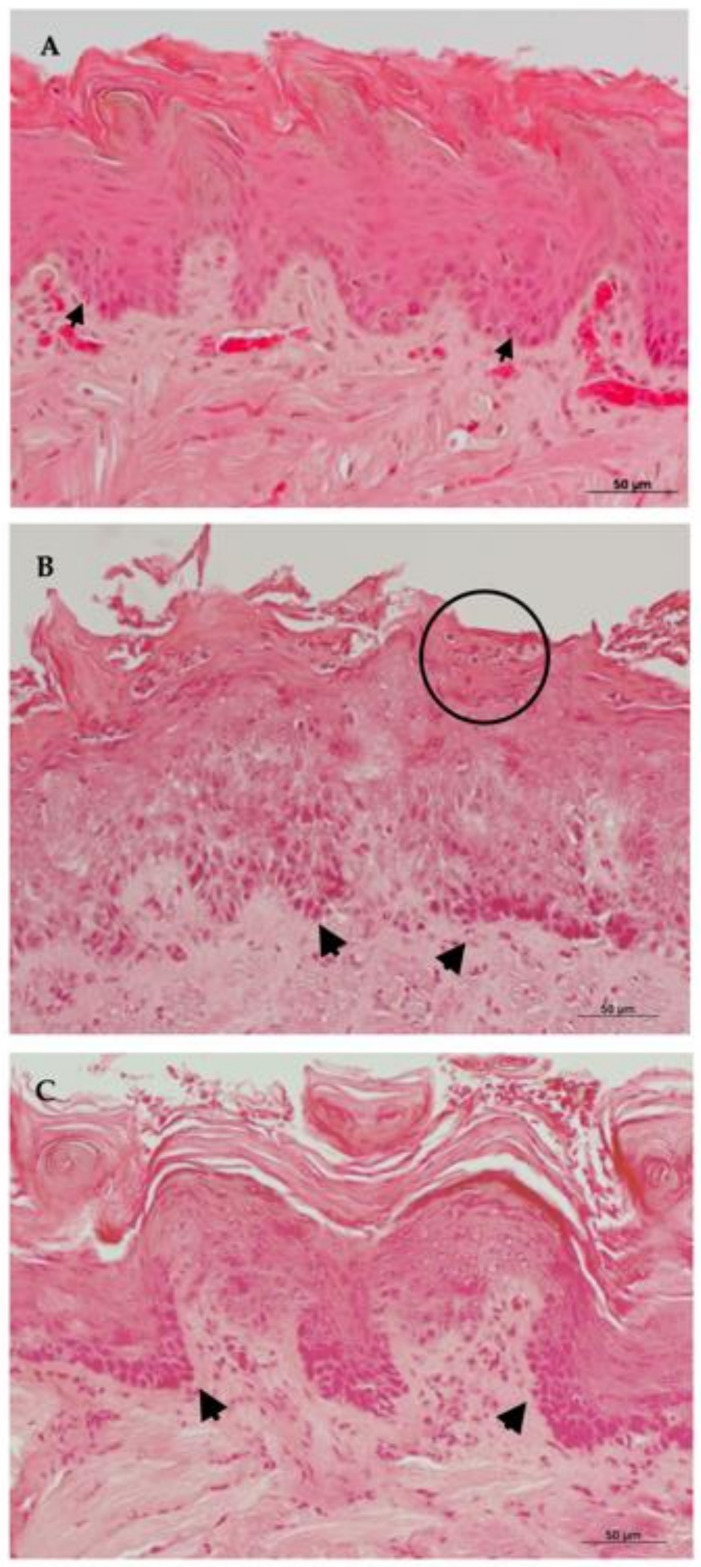
Histological sections from 24 h groups stained by Hematoxylin-Eosin (HE) in 200× magnification. Group treated with: (**A**) Nystatin; (**B**) ellagic acid complexed in HP-β-CD (EA/HP-β-CD) and (**C**) Control Group. Arrows show alterations on the basal layer. The circle highlights micro-abscess formation in the epithelium.

**Figure 5 molecules-26-00505-f005:**
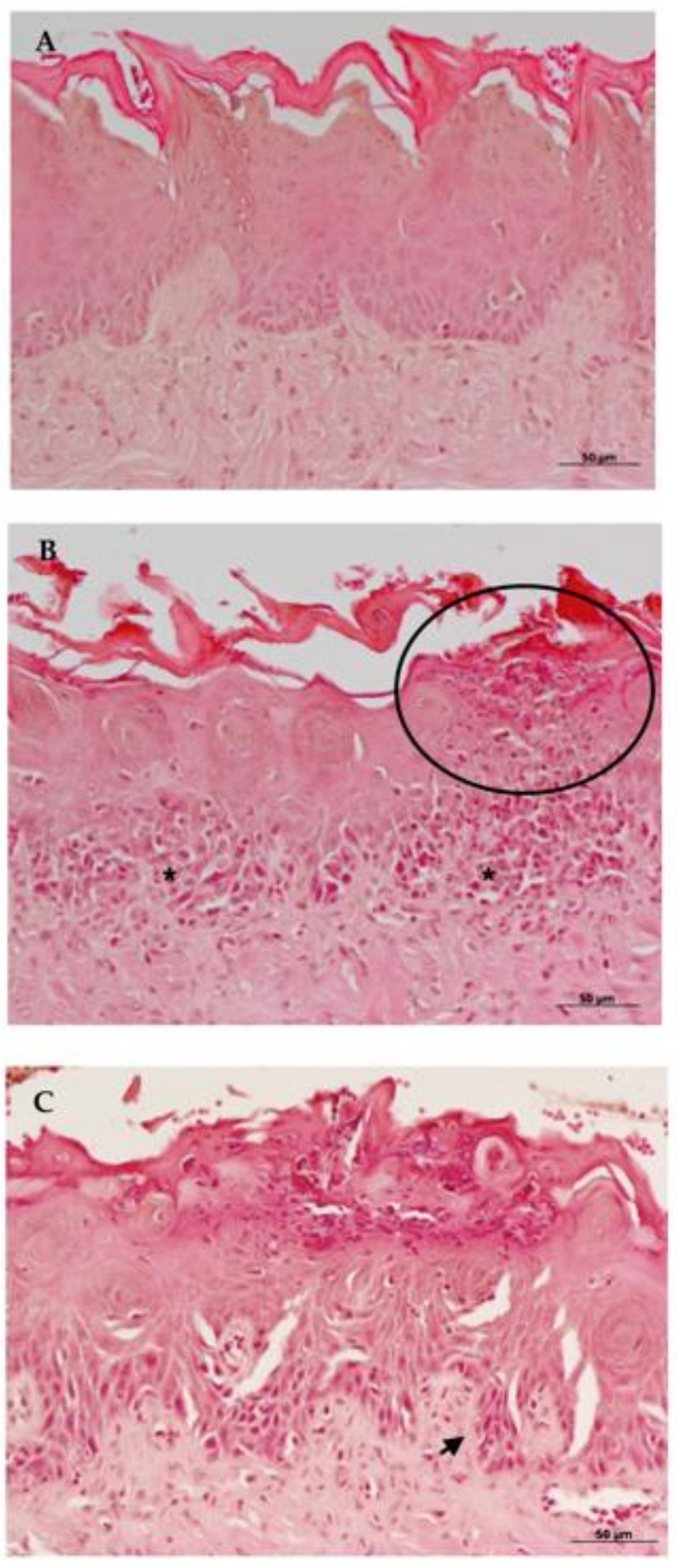
Histological sections from 96-h groups stained by HE in 200× magnification. Group treated with: (**A**) Nystatin, (**B**) ellagic acid complexed in HP-β-CD (EA/HP-β-CD), and (**C**) Control Group. Arrows show alterations on the basal layer. The circle highlights micro-abscess formation in the epithelium. (*) shows intense exocytosis in the basal layer of the epithelium.

**Figure 6 molecules-26-00505-f006:**
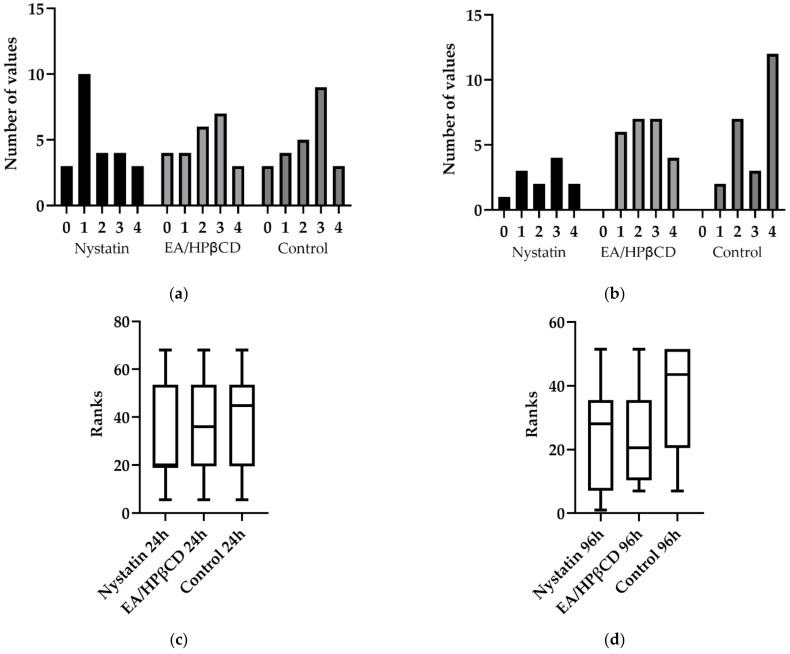
Frequency distribution of histological scores from groups: (**a**) 24 h; (**b**) 96 h; and ranks distribution and median from (**c**) 24 h and (**d**) 96 h.

**Figure 7 molecules-26-00505-f007:**
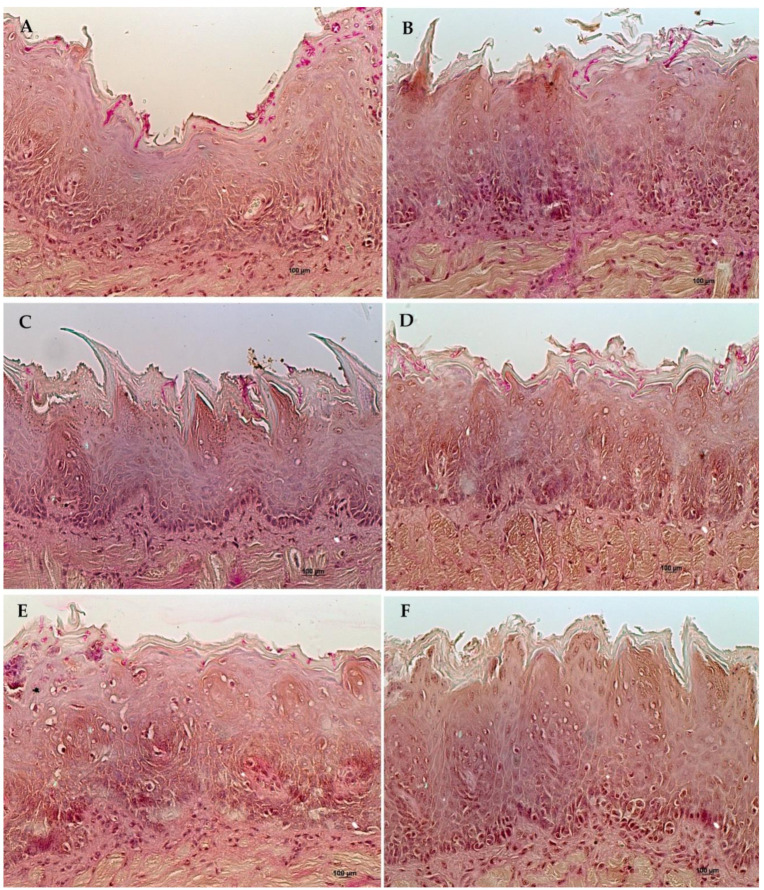
Histological sections stained by Periodic acid–Schiff (PAS) in 200× magnification. Group treated with Nystatin: (**A**) 24 h; (**B**) 96 h; group treated with ellagic acid complexed in HP-β-CD (EA/HP-β-CD) (**C**) 24 h; (**D**) 96 h and untreated control group (**E**) 24 h; (**F**) 96 h.

**Figure 8 molecules-26-00505-f008:**
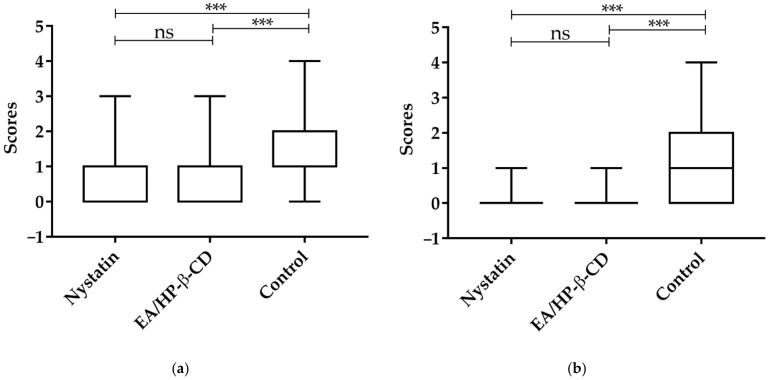
Scores (median values) of the histological analysis of *Candida* hyphae invading the epithelium after: (**a**) 24 h; (**b**) 96 h; from the last treatment. Statistical test was ANOVA followed by Kruskal–Wallis and multiple comparison Dunn’s test. *** *p* < 0.0001. ns—not significant.

**Figure 9 molecules-26-00505-f009:**
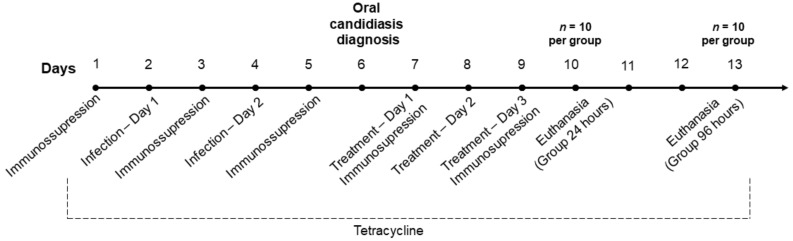
Timeline of the experimental procedure.

**Table 1 molecules-26-00505-t001:** Score of inflammatory epithelial alterations.

Score	Classification
0	No alteration
1	Hyperkeratosis or Hyperplasia
2	Hyperkeratosis + 2 inflammatory alterations
3	Hyperkeratosis + 3 inflammatory alterations including papillae rectification
4	Micro abscess + inflammatory alterations

## Data Availability

The data presented in this study are available on request from the corresponding author.

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
