# Peer review of "Ellagic Acid–Cyclodextrin Complexes for the Treatment of Oral Candidiasis"

_molecules, 2021, doi:10.3390/molecules26020505_

Round 1
Reviewer 1 Report
Journal: MDPI - Molecules
Manuscript
Title: " Ellagic Acid-Cyclodextrin Complexes for the Treatment of Oral Candidiasis"
Author(s): Aline da Graça Sampaio, Aline Vidal Lacerda Gontijo, Gabriela de Morais Gouvêa Lima, Maria Alcionéia Car-valho de Oliveira, Laura Soares Souto Lepesqueur and Cristiane Yumi Koga-Ito
Reviewer Comments to Author(s)
Recommendation: Major revisions
In this articles the authors researched on the potential use of ellagic acid-cyclodextrin complexes (EA/HP-β-CD) for the treatment of oral candidiasis. For this research work the effect of EA complexes on Candida albicans planktonic cells and biofilms, followed by in vitro and in vivo evaluation was assessed in fungal epithelial invasion in animal models. This is an interesting research report, however there are major concerns. The author(s) maybe should consider of the following:
- Introduction part: The introductory part has significant shortcomings in terms of state-of-the-art content. It has no references at all to cutting-edge technologies concerning the study and applications of ellagic acid. There are no references on the use, research, study and application of ellagic acid in cyclodextrin complexes. However, there is plenty of literature related to this subject, such as, V.D. Bulani et al., Inclusion complex of ellagic acid with β-cyclodextrin: Characterization and in vitro anti-inflammatory evaluation, Journal of Molecular Structure, Volume 1105, 5 February 2016, Pages 308-315, https://doi.org/10.1016/j.molstruc.2015.08.054 ; A.V.L. Contijo, Biopharmaceutical and antifungal properties of ellagic acid-cyclodextrin using an in vitro model of invasive candidiasis, Future Meicrobiology, 14(11) 2019 https://doi.org/10.2217/fmb-2019-0107 ;Savic et.al, The effect of complexation with cyclodextrins on the antioxidant and antimicrobial activity of ellagic acid, https://doi.org/10.1080/10837450.2018.1502318 ;Bulani, V.D., Kothavade, P.S., Nagmoti, D.M. et al. Characterisation and anti-inflammatory evaluation of the inclusion complex of ellagic acid with hydroxypropyl-β-cyclodextrin. J Incl Phenom Macrocycl Chem 82, 361–372 (2015). https://doi.org/10.1007/s10847-015-0498-7
Much more related research literature can be found by the auhtor(s) to set up a remarkable and extended state-of-the-art part in introduction. Moreover, there is an open access review article in MDPI-applied sciences (Formulation Strategies to Improve Oral Bioavailability of Ellagic Acid, https://doi.org/10.3390/app10103353) concerning oral bioavailability of ellagic acid in various formulations including cyclodextrins. In the review article reference 29 of this manuscript has been referenced and evaluated concerning Candida albicans infection. Maybe the authour(s) should consider that the previous research work should be stated in more detail and the differences and advantages of this research work should be outlined.
- Paragraph 2.3.1.: Cell type should be mentioned by the author(s) here, since the analytical part of Materials is at the end of the manuscript. The reader(s) should be aware of the cell type and cell viability assay at this part. Moreover, time details of the in vitro evaluation are not mentioned either in the text (here) or in the figures. Line 76 & 79: This type of presentation of the results is confusing. MIC is 25 μg/ml, 2 times is 50 μg/ml and so on. The author(s) are kindly advised to be more specific and detailed in the presentation of their results.
- Line 86: the author(s) state here that 250 μg/ml of EA (which by the way is the most cytotoxic concentration of all at 24 h) was selected according to the antibiofilm effect. However, only this concentration is mentioned to be evaluated in the 4.2 paragraph. Why the author(s) did not evaluated the rest EA concentrations? Moreover, as they state at line 88 the in vivo protocol was followed at 24 and 96 h. However, the in vitro viability was followed only at 24 h. Between in vitro and in vivo evaluation of complexed EA there is the gap of the 96 h evaluation. It is very important to provide the cell viability results at 96 h, since EA and complexed EA provided increased cytotoxicity at 24 h (or the lowest viability among all concentrations). The 250 μg/ml of EA and complexed EA provided 59% and 63% cell viability at 24 h. If cell viability will not be stabilized at a value above 50 % at 96 h (IC50 of EA), then the author(s) should explain why they used this concentration (250 μg/ml) that was the most cytotoxic.
- Discussion part: The analytical results of basal layer disorders (wherever observed) should be mentioned and visualized (maybe by arrows) in the histological sections of figure 4, in order to be easier for the reader to focus on these differences and the effectiveness of complexed EA.
- As mentioned before the timeline is an important characteristic in cytotoxicity evaluation. The author(s) have a timeline of 24 h for in vitro and 24 and 96 h for in vivo assays. in this paragraph (lines 194-213) the time period of the referenced studies should be provided for comparison reasons by the author(s).
- The author(s) are kindly requested to provide more information. The low cytotoxicity is 63% at 24 h at 250 μg/ml of complexed EA and the promising antibiofilm activity is not compared to any other concentration of complexed EA. Which is the cytotoxicity at 250 μg/ml of complexed EA at 96 h? The presentation of the results at 96 hours and comparative antibiofilm activity at varied concentrations is considered essential as it will verify the selection of the specific concentration (250 μg/ml) by the author(s) for the in vivo evaluation.
Author Response
Responses to reviewers’ comments
Reviewer #1:
Thank you for your valuable comments and the revision of our manuscript, “Ellagic acid-cyclodextrin complexes for the treatment of oral candidiasis”. All the comments have been taken into consideration and answered. The manuscript has been modified accordingly and all the incorporated information was highlighted in the correct version, with changes marked in red print.
“In this article the authors researched on the potential use of ellagic acid-cyclodextrin complexes (EA/HP-β-CD) for the treatment of oral candidiasis. For this research work the effect of EA complexes on Candida albicans planktonic cells and biofilms, followed by in vitro and in vivo evaluation was assessed in fungal epithelial invasion in animal models. This is an interesting research report, however there are major concerns. The author(s) maybe should consider of the following”:
- Introduction part: The introductory part has significant shortcomings in terms of state-of-the-art content. It has no references at all to cutting-edge technologies concerning the study and applications of ellagic acid. There are no references on the use, research, study and application of ellagic acid in cyclodextrin complexes. However, there is plenty of literature related to this subject, such as, V.D. Bulani et al., Inclusion complex of ellagic acid with β-cyclodextrin: Characterization and in vitro anti-inflammatory evaluation, Journal of Molecular Structure, Volume 1105, 5 February 2016, Pages 308-315, https://doi.org/10.1016/j.molstruc.2015.08.054 ; A.V.L. Contijo, Biopharmaceutical and antifungal properties of ellagic acid-cyclodextrin using an in vitro model of invasive candidiasis, Future Meicrobiology, 14(11) 2019 https://doi.org/10.2217/fmb-2019-0107 ;Savic et.al, The effect of complexation with cyclodextrins on the antioxidant and antimicrobial activity of ellagic acid, https://doi.org/10.1080/10837450.2018.1502318 ;Bulani, V.D., Kothavade, P.S., Nagmoti, D.M. et al. Characterisation and anti-inflammatory evaluation of the inclusion complex of ellagic acid with hydroxypropyl-β-cyclodextrin. J Incl Phenom Macrocycl Chem 82, 361–372 (2015). https://doi.org/10.1007/s10847-015-0498-7
Much more related research literature can be found by the auhtor(s) to set up a remarkable and extended state-of-the-art part in introduction. Moreover, there is an open access review article in MDPI-applied sciences (Formulation Strategies to Improve Oral Bioavailability of Ellagic Acid, https://doi.org/10.3390/app10103353) concerning oral bioavailability of ellagic acid in various formulations including cyclodextrins. In the review article reference 29 of this manuscript has been referenced and evaluated concerning Candida albicans infection. Maybe the authour(s) should consider that the previous research work should be stated in more detail and the differences and advantages of this research work should be outlined.
Thank you very much for your comments. As suggested, more details on the indicated studies involving the use and applicability of the ellagic acid and cyclodextrin complex were included in introduction section (page 2, line 51-70).
- Paragraph 2.3.1.: Cell type should be mentioned by the author(s) here, since the analytical part of Materials is at the end of the manuscript. The reader(s) should be aware of the cell type and cell viability assay at this part. Moreover, time details of the in vitro evaluation are not mentioned either in the text (here) or in the figures. Line 76 & 79: This type of presentation of the results is confusing. MIC is 25 μg/ml, 2 times is 50 μg/ml and so on. The author(s) are kindly advised to be more specific and detailed in the presentation of their results.
Thank you for your comments. As suggested, the information on cell type and cell viability assay was added to the Results section. The paragraph was rewritten to improve the presentation of results. The modified parts were highlighted (page 3, lines 90–98) and the subtitle of figure 2 (page 3, lines 102).
- Line 86: the author(s) state here that 250 μg/ml of EA (which by the way is the most cytotoxic concentration of all at 24 h) was selected according to the antibiofilm effect. However, only this concentration is mentioned to be evaluated in the 4.2 paragraph. Why the author(s) did not evaluated the rest EA concentrations?
As mentioned in the discussion (page 11, lines 208-213), the concentration 10 times MIC (250 µg/mL) was chosen, since biofilms are more resistant to the antimicrobial agents than planktonic cells. Thus, higher concentrations than MIC are usually necessary to be more effective against Candida biofilms, as previously described in the literature [1,2]. Therefore, concentrations lower or higher than 10 times MIC (250 µg/mL) would probably be, respectively, less effective and toxic or insoluble in culture medium. This information was added to the manuscript, since it was not really so clear.
Moreover, as they state at line 88 the in vivo protocol was followed at 24 and 96 h. However, the in vitro viability was followed only at 24 h. Between in vitro and in vivo evaluation of complexed EA there is the gap of the 96 h evaluation. It is very important to provide the cell viability results at 96 h, since EA and complexed EA provided increased cytotoxicity at 24 h (or the lowest viability among all concentrations). The 250 μg/ml of EA and complexed EA provided 59% and 63% cell viability at 24 h. If cell viability will not be stabilized at a value above 50 % at 96 h (IC50 of EA), then the author(s) should explain why they used this concentration (250 μg/ml) that was the most cytotoxic.
We understand these concerns of the reviewer about the cumulative toxicity of EA/HPβCD after repeated exposures over a long period, but we would like to clarify here that this time of 24 h and 96 h of the in vivo protocol was not the time of exposure to the drug, but the time when the euthanasia was performed after the last treatment (page 14, line 376 and 377).
It is worth mentioning that a volume of 100 μL of EA/HPβCD or nystatin is administered directly to the tongue of animals. Hence, most of the solution / suspension is probably swallowed and the contact time with this concentration was likely short. Therefore, this time of 24 h of exposure for cytotoxicity tests was not chosen to correlate with these times of euthanasia mice after the last treatment. This time was chosen according to the basic procedure described by International Standard (ISO 10993-5 - Biological evaluation of medical devices — Part 5: Tests for in vitro cytotoxicity) [3], to the vast majority of previous studies in the literature [4–6] and to previous works published by our group [7–9].
The MTT cytotoxicity test, first described in 1983 [10], is a classic, simple, ethical and fast methodology to predict the toxicity of a chemical compound following its exposure [11]. Nonetheless, it is not an ideal test to determine toxicity after repeated exposures over a long period, such as other methodologies described in [12]. Therefore, we added in the discussion of the manuscript that further studies should be carried out to determine the chronic toxicity of EA/HPβCD.
- Discussion part: The analytical results of basal layer disorders (wherever observed) should be mentioned and visualized (maybe by arrows) in the histological sections of figure 4, in order to be easier for the reader to focus on these differences and the effectiveness of complexed EA.
Thank you for the comments. As suggested, the figures of HE were identified appropriately. Basal layer disorders, that are already described on the text body, were highlighted on sections figures by arrows, as well as other important features were highlighted using other symbols.
For best visualization of sections’ tissue architecture, Figure 4 was divided in 2 new figures: Figure 4 (page 6, line 161) and Figure 5 (page 7, line 165).
In order to standardize group order on Figures, Figure 4, Figure 5 (new), and new Figure 8 group sequence have been modified.
- As mentioned before the timeline is an important characteristic in cytotoxicity evaluation. The author(s) have a timeline of 24 h for in vitro and 24 and 96 h for in vivo assays. in this paragraph (lines 194-213) the time period of the referenced studies should be provided for comparison reasons by the author(s).
All of these cited references 19 (page 11, line 233), 53 (page 11, line 240) and 54 (page 11, line 242) used 24 h of exposure time. In order to clarify possible doubts from readers, this information was added to the manuscript.
- The author(s) are kindly requested to provide more information. The low cytotoxicity is 63% at 24 h at 250 μg/ml of complexed EA and the promising antibiofilm activity is not compared to any other concentration of complexed EA. Which is the cytotoxicity at 250 μg/ml of complexed EA at 96 h? The presentation of the results at 96 hours and comparative antibiofilm activity at varied concentrations is considered essential as it will verify the selection of the specific concentration (250 μg/ml) by the author(s) for the in vivo evaluation.
The reasons for choosing the 24 h exposure time in cytotoxicity tests and the choice of 10 times MIC (250 μg/mL) for antibiofilm activity were explained in question 3.
References
[1] Rajendran, R.; Sherry, L.; Nile, C.J.; Sherriff, A.; Johnson, E.M.; Hanson, M.F.; Williams, C.; Munro, C.A.; Jones, B.J.; Ramage, G. Biofilm Formation Is a Risk Factor for Mortality in Patients with Candida Albicans Bloodstream Infection—Scotland, 2012–2013. Clin. Microbiol. Infect., 2016, 22, 87–93.
[2] Romera, D.; Aguilera-Correa, J.J.; Gadea, I.; Viñuela-Sandoval, L.; García-Rodríguez, J.; Esteban, J. Candida Auris: A Comparison between Planktonic and Biofilm Susceptibility to Antifungal Drugs. J. Med. Microbiol., 2019, 68, 1353–1358.
[3] International Organization for Standardization. ISO 10993-5 Biological Evaluation of Medical Devices -Part 5: Tests for in Vitro Cytotoxicity. 2009, 2006.
[4] Correa, R.M. dos S.; Mota, T.C.; Guimarães, A.C.; Bonfim, L.T.; Burbano, R.R.; Bahia, M. de O. Cytotoxic and Genotoxic Effects of Fluconazole on African Green Monkey Kidney (Vero) Cell Line. Biomed Res. Int., 2018, 2018, 1–7.
[5] Handayani, D.; Rasyid, W.; Rustini; Zainudin, E.N.; Hertiani, T. Cytotoxic Activity Screening of Fungal Extracts Derived from the West Sumatran Marine Sponge Haliclona Fascigera to Several Human Cell Lines: Hela, WiDr, T47D and Vero. J. Appl. Pharm. Sci., 2018, 8, 055–058.
[6] Uthaya Kumar, U.S.; Jothy, S.L.; Kavitha, N.; Chen, Y.; Kanwar, J.R.; Sasidharan, S. Genoprotection and Cytotoxicity of Cassia Surattensis Seed Extract on Vero Cell Evaluated by Comet and Cytotoxicity Assays. Proc. Natl. Acad. Sci. India Sect. B Biol. Sci., 2018, 88, 313–320.
[7] Brighenti, F.L.; Salvador, M.J.; Gontijo, A.V.L.; Delbem, A.C.B.; Delbem, Á.C.B.; Soares, C.P.; De Oliveira, M.A.C.; Girondi, C.M.; Koga-Ito, C.Y. Plant Extracts: Initial Screening, Identification of Bioactive Compounds and Effect against Candida Albicans Biofilms. Future Microbiol., 2017, 12, 15–27.
[8] Oliveira, M.A.C.; Borges, A.C.; Brighenti, F.L.; Salvador, M.J.; Gontijo, A.V.L.; Koga-Ito, C.Y. Cymbopogon Citratus Essential Oil: Effect on Polymicrobial Caries-Related Biofilm with Low Cytotoxicity. Braz. Oral Res., 2017, 31, 1–12.
[9] Teodoro, G.R.; Gontijo, A.V.L.; Salvador, M.J.; Tanaka, M.H.; Brighenti, F.L.; Delbem, A.C.B.; Delbem, Á.C.B.; Koga-Ito, C.Y. Effects of Acetone Fraction From Buchenavia Tomentosa Aqueous Extract and Gallic Acid on Candida Albicans Biofilms and Virulence Factors. Front. Microbiol., 2018, 9, 1–10.
[10] Mosmann, T. Rapid Colorimetric Assay for Cellular Growth and Survival: Application to Proliferation and Cytotoxicity Assays. J. Immunol. Methods, 1983, 65, 55–63.
[11] Fotakis, G.; Timbrell, J.A. In Vitro Cytotoxicity Assays: Comparison of LDH, Neutral Red, MTT and Protein Assay in Hepatoma Cell Lines Following Exposure to Cadmium Chloride. Toxicol. Lett., 2006, 160, 171–177.
[12] Nehoff, H.; Parayath, N.N.; Taurin, S.; Greish, K. In Vivo Evaluation of Acute and Chronic Nanotoxicity. In Handbook of Nanotoxicology, Nanomedicine and Stem Cell Use in Toxicology; John Wiley & Sons, Ltd: Chichester, UK, 2014; pp. 65–86.

Reviewer 2 Report
The topic of this manuscript is interesting. Although similar researches have been reported before, as the authors provided in vivo data, the reviewer suggests the acceptance of this manuscript after some minor amendments.
(1) The reviewer has more than 20 year experiences with cyclodextrin. According to his own knowledge, beta-cyclodextrin or hydroxypropyl-beta-cyclodextrin are not the best vehicle to solubilize ellagic acid. Have the authors attempted phase-solubility study? Gamma-cyclodextrin and hydroxypropyl-gamma-cyclodextrin may be better. Anyway, this is just a minor point.
(2) The statistical protocol is wrong. From comparison between two groups, t-test and its non-parametric equivalent should be applied. Unfortunately, the authors used ANOVA and its non-parametric equivalent.
(3) The authors should discuss how to translate their findings in clinical setting.
Author Response
Responses to reviewers’ comments
Reviewer #2:
Thank you for your valuable comments and the revision of our manuscript, “Ellagic acid-cyclodextrin complexes for the treatment of oral candidiasis”. All the comments have been taken into consideration and answered. The manuscript has been modified accordingly and all the incorporated information was highlighted in the correct version, with changes marked in red print.
The topic of this manuscript is interesting. Although similar researches have been reported before, as the authors provided in vivo data, the reviewer suggests the acceptance of this manuscript after some minor amendments.
- The reviewer has more than 20 year experiences with cyclodextrin. According to his own knowledge, beta-cyclodextrin or hydroxypropyl-beta-cyclodextrin are not the best vehicle to solubilize ellagic acid. Have the authors attempted phase-solubility study? Gamma-cyclodextrin and hydroxypropyl-gamma-cyclodextrin may be better. Anyway, this is just a minor point.
Thank you the very much for the valuable suggestion. They will be certainly very interesting to our future studies.
We evaluate the solubility diagram of EA (ellagic acid) in hydroxypropyl-beta-cyclodextrin (HP-β-CD) and methyl-beta-cyclodextrin (M-β-CD) as published in Gontijo et al (2019) [1] and we observed best result in HP-β-CD cyclodextrin in NaOH 0.1M, with solubility curve of R² = 0.9116.
- The statistical protocol is wrong. From comparison between two groups, t-test and its non-parametric equivalent should be applied. Unfortunately, the authors used ANOVA and its non-parametric equivalent.
Sorry, the suggestion was not clear. All the statistic test were analyzed with comparison between 3 groups: 1-biofilm test (page 3, line 84): (a) pure-EA group, (b) EA/HP-β-CD group and (c) control group; 2- number viable fungal cell recovered (page 4, line 118): (a) nystatin group, (b) EA/ HP-β-CD group and (c) control group; 3- analysis of Candida hyphae invading the epithelium (page 9, line 183): (a) nystatin group, (b) EA/ HP-β-CD group and (c) control group.
- The authors should discuss how to translate their findings in clinical setting.
Thank you for the comments. The requested information was added to the Discussion (page 12, lines 297-302).
References
[1] Gontijo, A.V.; G Sampaio, A. da; Koga-Ito, C.Y.; Salvador, M.J. Biopharmaceutical and Antifungal Properties of Ellagic Acid-Cyclodextrin Using an in Vitro Model of Invasive Candidiasis. Future Microbiol., 2019, 14, 957–967.

Round 2
Reviewer 1 Report
Journal: MDPI - Molecules
Manuscript
Title: " Ellagic Acid-Cyclodextrin Complexes for the Treatment of Oral Candidiasis"
Author(s): Aline da Graça Sampaio, Aline Vidal Lacerda Gontijo, Gabriela de Morais Gouvêa Lima, Maria Alcionéia Car-valho de Oliveira, Laura Soares Souto Lepesqueur and Cristiane Yumi Koga-Ito
Reviewer Comments to Author(s)
Recommendation: Accept
In this articles the authors researched on the potential use of ellagic acid-cyclodextrin complexes (EA/HP-β-CD) for the treatment of oral candidiasis. For this research work the effect of EA complexes on Candida albicans planktonic cells and biofilms, followed by in vitro and in vivo evaluation was assessed in fungal epithelial invasion in animal models.
After a detailed evaluation of the manuscript after revision all the issues and questions have been addressed by the author(s) and the manuscript can be accepted for publication.